# The Brain Dynamics of Syllable Duration and Semantic Predictability in Spanish

**DOI:** 10.3390/brainsci12040458

**Published:** 2022-03-29

**Authors:** Mercedes Muñetón-Ayala, Manuel De Vega, John Fredy Ochoa-Gómez, David Beltrán

**Affiliations:** 1Programa de Filología Hispánica, Facultad de Comunicaciones y Filología, Universidad de Antioquia, Calle 70 N° 52-21, Medellín 050010, Colombia; 2Instituto Universitario de Neurociencia, Universidad de la Laguna, 38200 Tenerife, Spain; mdevega@ull.edu.es (M.D.V.); dbeltran@ull.edu.es (D.B.); 3Programa de Bioingeniería, Facultad de Ingeniería, Universidad de Antioquia, Medellín 050010, Colombia; john.ochoa@udea.edu.co; 4Laboratorio de Neurofisiología, GRUNECO-GNA, Universidad de Antioquia, Medellín 050010, Colombia; 5Departamento de Psicología Básica, Universidad Nacional de Educación a Distancia, 28040 Madrid, Spain

**Keywords:** semantic predictability, ERPs, Spanish prosody, N400, P800

## Abstract

This study examines the neural dynamics underlying the prosodic (duration) and the semantic dimensions in Spanish sentence perception. Specifically, we investigated whether adult listeners are aware of changes in the duration of a pretonic syllable of words that were either semantically predictable or unpredictable from the preceding sentential context. Participants listened to the sentences with instructions to make prosodic or semantic judgments, while their EEG was recorded. For both accuracy and RTs, the results revealed an interaction between duration and semantics. ERP analysis exposed an interactive effect between task, duration and semantic, showing that both processes share neural resources. There was an enhanced negativity on semantic process (N400) and an extended positivity associated with anomalous duration. Source estimation for the N400 component revealed activations in the frontal gyrus for the semantic contrast and in the parietal postcentral gyrus for duration contrast in the metric task, while activation in the sub-lobar insula was observed for the semantic task. The source of the late positive components was located on posterior cingulate. Hence, the ERP data support the idea that semantic and prosodic levels are processed by similar neural networks, and the two linguistic dimensions influence each other during the decision-making stage in the metric and semantic judgment tasks.

## 1. Introduction

Prosody is a complex aspect of communicative speech act that requires the successful integration of multiple acoustic parameters, such as fundamental frequency (F0), duration and intensity, whose perceptual correlates are pitch, timing and loudness, respectively, all of which contribute to the perception of the suprasegmental structure of sentences. In several studies related to Spanish language, the most studied parameter has been F0, which, according to some authors, plays the principal role in marking prominent syllables in speech [1]. Yet, some recent studies have also shown the importance of duration and intensity [2,3,4,5,6]. The present study specifically focused on duration, defined as the time taken to utter any part of the speech signal [4], for instance, of the syllable. 

In general, languages have been described to fall into one of two rhythmic, mutually exclusive categories based on isochrony, depending on whether they have equal intervals of time for syllables or stress, called syllable-timed languages and stress-timed languages, respectively [7]. The former are languages with stable syllable duration, such as French, Italian or Spanish, and the latter are languages with stable duration for interstress interval, such as English, Dutch or Arabic [8,9]. However, some studies do not support the isochrony principle to differentiate languages. For example, in a cross-linguistic study (English, Thai, Spanish, Italian and Greek) in which informants had to read a passage from a contemporary novel translated to their own language, Dauer [10] showed that rhythmic grouping occurs more or less regularly not only in English (stress-timed language) but also in Spanish (syllable-timed language). Additionally, Dorta and Mora [11] analyzed the rhythmic characteristics of two dialectal varieties of spoken Spanish in the Canary Islands and Venezuela. They focused on the syllable as the rhythmic unity, studying the timing behavior of the syllabic nucleus, according to different segmental and suprasegmental factors. The results did not support the hypothesis of syllable-timed rhythm in Spanish because stressed syllables were longer than unstressed syllables. The same results have been found by other researchers in Spanish language studies [12,13,14,15]. 

Additionally, it is well known that vowel or syllabic duration has a phonological and a phonetic dimension with specific functional consequences. Its phonological dimension allows the distinction of one word from another in some languages. For example, in Japanese, there are pairs such as /isso/ (“rather”) vs. /isso:/ (“more”); in Finnish, there are triplets such as /tule/ (“come”) vs. /tule:/ (“comes”) vs. /tu:le/ (“it blow”) [4]; meanwhile, the phonetic dimension does not change the meaning of a word as in Spanish. However, as mentioned above, in this language we can differentiate between long (tonic syllable) and short vowels (adjacent syllables [16] (p. 55)). These characteristics make Spanish an interesting language for studying the influence of duration on the perception of sentences. 

### 1.1. Duration Approaches

The relevance of duration features in prosody has been investigated throughout different approaches, such as neuropsychological [17,18] and electrophysiological [19,20,21,22,23]. Neuropsychological studies have shown that duration features in prosody can be used to distinguish lexical items both in speech perception [17] and production [18]. The results agree in pointing out that temporal information, such as syllable duration, is processed in the left hemisphere. For example, Van Lancker and Sidtis [17] found that left-hemisphere-damaged patients (LHD) and right-hemisphere-damaged patients (RHD) utilize acoustic cues differently to make a prosody judgment task. Namely, syllable duration variability was the principal cue used by RHD patients, while F0 variability was used by LHD patients, indicating that activation of the left hemisphere and right hemisphere is related to the durational cue and the F0 cue, respectively. In the same vein, Yang and Van Lancker [18] investigated the production of idiomatic and literal expression in left- and right-hemisphere-damaged and in normal Korean speakers. The major finding was that distinguishing the two types of expressions relied on F0 changes in speakers with LHD, while duration was the main cue in speakers with RHD, confirming that the left hemisphere is specialized to process temporal cues, and the right hemisphere is specialized to process pitch cues. However, no conclusive information is available about this subject. A different study showed that the linguistic prosody process is as bilateral as the emotional prosody [24] or that prosody is lateralized to the right irrespective of the communicative function [25]. 

The event-related potential technique (ERP) has found a signature pattern of brain activity to index semantic or phonetic congruency during language comprehension [26]. N400 is one of the most important components in the research of language, with a negative polarity that peaks around 400 ms after the word onset. Specifically, it is a useful signature pattern for addressing questions on the integration of prosodic information in auditory processing [20], providing a measure of the time course of prosodic integration in semantic [22,23] and syntactic processes [20,27]. In these studies, researchers manipulated suprasegmental characteristics using words with incorrect lexical/metrical stress patterns [21,28,29,30], words with correct but unexpected stress patterns [22,23,31,32] or unexpected stress patterns in words or pseudowords [33]. The results indicate that subjects are aware of the stimuli changes, and their brain is sensitive to the subtle violation of rhythmical structure, which can influence the semantic encoding. For instance, Böcker et al. [31] investigated how listeners process words starting with the alternation of strong and weak syllables in a stress-timed language, such as Dutch. They found that initially weak words, as compared to initially strong words, elicited a negative brain response, probably related to stress discrimination, peaking at around 325 ms post-stimulus onset and maximum at the frontocentral scalp, likely reflecting the modulation of an anterior N400 component. An ERP study by Bohn et al. [32] that manipulated the German rhythm (alternation of stressed and unstressed syllable) in auditorily presented words found that where irregular but possible meter words involve semantic cost (i.e., enhanced N400 to unexpected stress change), those with regular meter do not. This suggests that regular meter is required to avoid additional cost in the semantic processing of words. 

Furthermore, other studies that have manipulated syllable duration have also found modulations of both the N400 and a late positive component (LPC). For example, Magne et al. [21] used a design to examine the relationship between semantic and prosodic processing in spoken French. In this case, the prosodic violation was realized as a lengthened instance of the pretonic syllables (the second one) of the critical trisyllabic word. The result showed the on-line processing of the metric structure of the words. They found an N400-like negativity and a LPC (P600) in the prosodic judgment task but only the N400 effect in the semantic judgment task for the incongruent lengthening. This suggested the automaticity of metrical structure processing and demonstrated that violations of a word’s metric structure may hinder lexical access and word comprehension. 

Similar LPC effects were reported by Astesano et al. [19]. They used semantically congruous and incongruous sentences, and sentences whose prosody matched or mismatched its syntactic form, by cross-shifting sentence beginning and ending. Therefore, they created four conditions: (1) both semantically and prosodically congruous; (2) semantically congruous and prosodically incongruous; (3) semantically incongruous and prosodically congruous; and finally, (4) both semantically and prosodically incongruous. Regarding prosody, they found a late positive component (P800) associated with prosodic mismatch. The late positivity was mediated by the task demand because it only emerged when prosody was in task focus. In the same vein, Paulmann et al. [34] compared the linguistic and emotional functions of prosody. The objective was to analyze whether the two prosodic functions engage a similar time course or not. To this aim, they merged a prosodically neutral head of a sentence to a second half of a sentence that differed in emotional and/or linguistic prosody. Consequently, the study consisted of: an emotional task in which participants judged whether the sentence that they had just heard was spoken in a neutral tone of voice or not; and a linguistic task in which participants decided whether the sentence was a declarative sentence or not. As was expected, the results reported a prosodic expectancy positivity irrespective of the task, but the latency for the linguistics prosody effect was later (~620 ms) than the latency for the emotional prosody violation (~470 ms). 

In general, these ERP findings reflect the influence of prosody on comprehension taking into account its linguistic functions, such as lexical access/integration [21,35] or judgment of the sentence modality [19]. 

### 1.2. The Present Study

The aim of the current study was to investigate the on-line processing of semantic and prosodic information, manipulating semantic predictability and syllable duration in a similar way as in the study reported by Magne et al. [21]. Yet, a crucial difference was that Magne et al.’s study was performed in French, a fixed-accent language, whereas this study was in Spanish, a free-accent language. So, in Spanish, words can be stressed on the last, the penultimate or the antepenultimate syllable (in very rare cases, it can come on the fourth last syllable in compound words). According to Quilis [36] (pp. 333–336), 79.5% of the words are stressed on the penultimate syllable. For this reason, we used this group of words in the present study, marking a clear difference between both studies. Thus, we manipulated the first syllable, and there is evidence that this syllable plays a major role and gives more information about the word than other syllables [37,38,39,40]. 

Another difference between both studies was the manipulation of the semantic dimension. Magne et al. [21] created the incongruent condition by replacing a congruent word by a word that was nonsensical in the sentence (low cloze probability). Meanwhile, in the present study, the semantic manipulation was subtler; both the predictable and unpredictable words were sensible in the sentence context. In this way, detecting a semantic anomaly requires a deeper semantic processing. 

On the other hand, note that the duration of the sound (syllable) is set relative to another sound [4]. Therefore, it is very important to consider how much the duration of two syllables must differ to produce a detectable perceptual difference. Pamies et al. [41] investigated the just noticeable differences (JND) for segment duration in Spanish language. They reported that an amount of 33.33% is sufficient to cause a difference in the perception of duration. In general, the majority of JNDs in Spanish are in the prepositional syntagma and between pretonic and tonic syllables [42,43]. Most of the time, the tonic syllable is larger than the pretonic one. For these reasons, we manipulated the duration in the prepositional syntagma, eliminating differences between these two adjacent syllables in our stimuli.

In sum, the aim of the present study was to examine the processing of prosodic and semantic information in spoken Spanish using event-related potentials (ERP) measurements and brain source localization. Specifically, we aimed to determine whether the prosodic (duration feature) and semantic aspects of spoken language are processed independently by the brain or whether they share neural resources. This process must be based on the statistics results. A statistical interaction between duration and semantics would indicate that both processes share brain activity, while independent statistical effects would be related to differentiated neural processes. An important manipulation was the orienting task, that is, the participants were asked to make semantic judgments in half of the trials and metric judgments in the remaining ones. The task allows researchers to examine whether semantic and duration features are processed implicitly (e.g., semantic effects under metric judgment task) or only when they are explicitly requested for the task. Based on the finding indicating difficulties in word recognition with an incorrect stress pattern, one might expect the inadequate prosodic pattern to lead to difficulties in lexical processing and integration. One prediction is that the prosodic manipulation will result in an early negative component (N400) and a late positive component (P800) independently of the judgment task, which is under explicit and implicit instructions. 

Relating to the source localization, it is not feasible to make specific predictions, given that, as far as our knowledge goes, studies that address the relationship between prosody and semantics have not located its source. Nonetheless, as the N400 is a component associated with semantic processing, a general prediction is that semantic-driven differences in this time window should be source located in the left hemisphere [44], regardless of the task. Likewise, sources for the P800 could locate in the right hemisphere, given it is more frequently associated with prosodic processing [45]. For this reason, this work has investigated the localization of the source in the brain.

## 2. Method

### 2.1. Participants

A total of 26 Colombian undergraduate students of psychology, medicine and philology participated in this experiment (12 females, mean age 20 years old, range 18–24). All were neurologically healthy, right-handed native speakers of Spanish and had normal or corrected-to-normal vision and normal hearing. The data of 6 participants had to be removed from the analysis because they contained excessive motion artifacts, leaving a final sample of 20 participants (12 females). The study was approved by Comité de Ética en Investigación Área de Ciencias Sociales Humanidades y Artes (CEI-CSHA) (35MM-2019), Universidad de Antioquia. All participants gave informed consent before the experiment was carried out, according to the Declaration of Helsinki. 

### 2.2. Design and Material 

The experimental design included three within-participant factors: × 2 Judgment task (metric vs. semantic), × 2 Duration congruence (congruous vs. incongruous), × 2 Semantic predictability (predictable vs. unpredictable). 

A total of 224 semantically predictable sentences (SP+) were constructed in Spanish, all of them ending with a trisyllable noun, with stress on the penultimate syllable. Semantically unpredictable sentences (SP−) were created by replacing the final predictable word. So, there were a total of 448 sentences. All words were matching in frequency (*t* (446) = 0.948, *p* = 0.334); the mean semantic predictability was 20.40 (SD 47.81) and 16.15 per million (SD 47.08) for SP+ and SP− words, respectively, using the ESPAL (https://www.bcbl.eu/databases/espal/, accessed on 6 December 2020), and both predictable and unpredictable sentences were semantically sensible. 

Next, the lengthening of the pretonic syllables belonging to the last word was modified in all sentences. However, assuming that syllable duration is not a phonological feature in Spanish, it is important to control the lengthening of the syllable in order to be ecological. For that, we took different characteristics into account. First, the JND between pretonic and tonic syllable disappears in incongruous duration condition in such a way that, most of the time, the tonic and the pretonic syllable have the same duration. Neither in incongruous duration (DC−) nor congruous duration (DC+) was the pretonic syllable longer than the tonic syllable (examples of the material are shown in Table 1). In sum, the sentences were sorted into four conditions: (1) congruous duration within predictable sentences (DC+ and SP+), (2) congruous duration within unpredictable sentences (DC+ and SP−), (3) incongruous duration within predictable sentences (DC− SP+) and (4) incongruous duration within unpredictable sentences (DC− and SP−), each one comprising 224 different sentences. 

Each subject listened to 224 different sentences in total (112 for semantic task and the rest for metric task). Each task comprised two blocks of trials, each including 14 sentences for each of the four conditions previously defined (DC+ and SP+; DC+ and SP−; DC− and SP+; DC− and SP−). The sentences within each block were presented at random. Thus, each sentence was presented in each experimental condition across the subjects with no repetition within the subjects. The order of presentation of the task and the blocks were counterbalanced across subjects in order to eliminate any effect of order of presentation on behavioral or ERPs data. 

### 2.3. Speech Signal

The recording of the sentences was conducted in the soundproof booth belonging to the phonetic laboratory of the University of Antioquia by a native female speaker of standard Colombian Spanish. The acquisition was conducted using the Goldwave software (4.25) at a sampling frequency of 44.1 KHz. All sentences were spoken in declarative mode, ending in a descendent intonation, as is typical in Spanish language. The intensity was normalized by mean value of intensity from each sentence before modifying the vowel duration. It was increased using the PSOLA facility of the PRAAT software [46] to modify the duration signal without modifying its timbre or frequency (Table 2). Paired t-tests were conducted to assess the differences in total word length between the congruous duration and incongruous duration. Results revealed that there were significant differences in both predictable (*t* 241 = −76.12, *p* < 0.001) and unpredictable words (*t* 241 = −40.99, *p* < 0.001). 

### 2.4. Procedure 

During the experiment, participants were seated in a comfortable chair in a soundproof room at an approximate distance between 75 and 85 cm from the screen with instructions to remain relaxed and to minimize the blinking during experiment when they listen to the speech. After receiving the instructions, participants were given 10 practice trials, followed by four blocks with 56 attempts each that mixed the conditions in a random order. In two blocks, participants were asked to pay attention only to the semantic in order to decide whether the last word was predictable or not (semantic task). In the other blocks, the participants were asked to pay attention to the metric features of the stimuli (syllabic duration); specifically, they were asked to decide whether the last word of each sentence was well pronounced or not (metric task). The different tasks were given in an alternating order. Participants responded by pressing the left or the right button of the gamepad. Half of the participants pressed the left button to judge the sentence as correct and the right button to judge a sentence as incorrect. The other half followed a reversed button assignment. The whole experiment, including the set-up of the electroencephalographic electrodes (EEG), took about 70 min. 

Each trial consisted of a sentence presented aurally through headphones at a 65 dB sound pressure level in a pseudorandom order. An experimental trial was carried out as follows: a white screen was presented for 1000 ms; then a black fixation cross that reminded the participants to avoid unnecessary eye movements was presented in the middle of the white screen for 1000 ms; and 1000 ms after the presentation of the cross, the subjects listened to the sentence. Participants were instructed to respond as quickly and accurately as they could. After each sentence presentation, the participant had a maximum response time of 3000 ms. All the events in the trial were controlled by means of E-prime software (version 2.1; Psychology Software Tools). 

### 2.5. ERP: Recording, Preprocessing and Data Analyses 

EEG and electrooculographic (EOG) signals were recorded using Ag/AgCl electrodes mounted in elastic Quick-Caps (Compumedics). The EOG signal was measured from two bipolar channels: one from two electrodes placed at the outer canthus of each eye and the other from two electrodes above and below the left eye. The EEG signal was recorded from 60 electrodes arranged according to the standard 10–20 system and also on the left and right mastoids (M1/M2). All EEG electrodes were referenced online to an electrode at vertex and re-referenced offline to the average of the mastoid electrodes (M1 and M2). EEG and EOG signals were amplified at a 500 Hz sampling rate using Synamp2 amplifier (Neuroscan; Compumedics) with high- and low-pass filters set at 0.05 and 100 Hz, respectively. EEG electrode impedance was kept at <5 kΩ. EEG data preprocessing and analysis were conducted using Fieldtrip Toolbox [47]. Trials with drifting or large movement artifacts were removed by visual inspection before analysis. Independent component analysis was applied to the data to remove the effects of blinks and eye movements. The remaining trials with EEG voltages exceeding 70 μV measured from peak to peak at any channel were also removed.

The EEG data were time locked to the last word onset to analyze the effects of the semantic and the metric dimensions of the stimuli. EEG segments for ERP computations were obtained by epoching from 200 ms before to 1200 ms after the stimulus onset. ERPs to correct responses were computed by averaging EEG segments from the same condition and participant, and using the 200 ms pre-stimulus onset as baseline. 

To statistically explore the time course of differences for the whole experimental design, we averaged the ERP amplitudes for successive 10 ms time windows, starting from stimulus onset. Next, the time windows (142 in total) were analyzed using repeated-measure ANOVA with Task (metric vs. semantic), Duration (DC+ vs. DC−) and Semantic predictability (SP+ vs. SP−) as within-subject conditions. Furthermore, to increase sensitivity for spatial variation of the effects, we included two additional within-subject factors: Hemisphere (left vs. right) × Region (frontal vs. central vs. parietal). The ROIs forming these spatial factors were grouped and averaged as follows: left frontal (F3, F5, FC3); right frontal (F4, F6, FC4); left central (C3, C5, CP3); right central (C4, C6, CP4); and left parietal (P3, P5, PO3) and right parietal (P4, P6, PO4). Finally, to somehow control for false positives due to the multiple analyses, we adopted the criteria of only the considered reliable effects that reached significance in five or more consecutive time windows. The aim of this exploratory analysis was to identify the interaction effects with Task. These ANOVAs were conducted on the mean of the amplitude for all the consecutive time windows that led to the initial detection of the Task–interaction effect. As will be described in the Results section, the exploratory analyses identified two time periods in which there were reliable interactions with Task, which partially overlapped with the development of N400 (300–390 ms) and LPC (1000–1200) components. 

To estimate the cortical regions involved in the pair-wise effects detected at surface, we ran brain sources analyses for the two time windows uncovered from the exploratory ERP analysis: 300–390 and 1000–1200 ms after critical word onset. The local auto-regressive average (LAURA; [48]) distributed source estimation approach was applied using Cartool software (http://brainmapping.unige.ch/Cartool.php) (accessed on 10 July 2018). A realistic head model based on a standard MRI (MNI brain from the Montreal Neurological Institute) was used by applying the SMAC transformation method, which transforms the MRI to a best fitting sphere. Next, 4026 points were defined in regular distances within the gray matter (6 × 6 × 6 mm), and a 3-shell spherical lead field was calculated for this set of solution points and for the 60 scalp electrodes. From this lead field, current density magnitudes (microampere per cubic meter) at each solution point were calculated per subject and performance condition and submitted to statistical analyses using paired *t*-tests. Next, source t-test maps were computed for the pair-wise comparisons that were significant in the above ERP analysis. Then, the regions of interest (ROIs) were defined following the solution points showing the maximal statistical differences above the threshold *t*-value ± 3.5 (*p* < 0.001). To estimate the Brodmann areas that best fit the LAURA results, the location of each maximum *t*-value (positive or negative) was compared with the location of the Brodmann areas in Talairach space [49]. 

## 3. Results

### 3.1. Behavioral Results

We performed a Task (metric vs. semantic) × Duration congruence (DC+ vs. DC−) × Semantic predictability (SP+ vs. SP−) repeated measure ANOVAs on transformed percentages of accuracy (Figure 1) and reaction times (Figure 2), including correct trials. Concerning accuracy, there was a significant duration by semantic interaction F(1, 19) = 11.08, *p* < 0.004, η_p_^2^ = 0.37. The pair-wise comparison revealed that correct answers in semantic conditions were influenced by the duration levels. During DC+, participants had more correct answers in SP+ than in SP− (*p* < 0.001). In the same vein, correct answers in the duration contrast were influenced by semantic levels; that is, the performance was better in DC+ than in DC− but only in semantically predictable cases, SP+ (*p* < 0.001). 

For reaction times, the interaction between duration and semantic also reached significance F(1, 19) = 4.59, *p* < 0.045, η_p_^2^ = 0.20. Taking into account the semantic contrast, the results showed that unpredictable sentences demand more time than predictable sentences both in congruous (*p* < 0.001) and in incongruous duration (*p* < 0.001). Taking into account the duration contrast, the results revealed that incongruous duration requires more time than congruous duration both in predictable (*p* < 0.001) and unpredictable (*p* < 0.002) sentences processed. Overall, the accuracy and reaction time results indicate that, regardless of the task, participants were sensitive to violations on both the metric and semantic level, and specifically, to the interaction between them. 

### 3.2. Electrophysiological Results

As noted above, the exploratory analysis of the EEG data showed reliable interactions with task in two time periods: from 300 to 390 ms and from 1000 to 1200 ms after the stimulus onset. These effects covered part of longer N400 (Figure 3 and Figure 4) and LPC (Figure 5) components. Next, we describe the results for the follow-up ANOVAs conducted on the mean amplitudes in these time windows.

#### 3.2.1. First Time Window (300–390)

The visual inspection of the waveforms (Figure 3 and Figure 4) reflects that the first time window, extending from 330 to 390 ms after last sentence word onset, coincides with an N400-like component. In this time window, there was the main effect of duration (F(1, 19) = 20.80, *p* < 0.001, η_p_^2^ = 0.52) where DC− had larger amplitude than CD+. The main effect was qualified by a significant interaction between Task, Duration and Semantic (F(1, 19) = 4.44, *p* < 0.049, η_p_^2^ = 0.19). The follow-up contrast revealed that, in the metric task, there was a semantic effect (DC+ SP+ vs. DC+ SP−) and a duration effect (DC+ SP+ vs. DC− SP+), as shown in Table 3. Concerning the semantic effect, unpredictable sentences elicited a more negative waveform (N400) than predictable sentences in the correct duration condition (see Figure 3B). Concerning the duration effect, incongruous duration elicited a more negative waveform than congruous duration in predictable sentences (see Figure 3C). By contrast, in the semantic task, the duration effect only happened for the unpredictable sentences (DC+ SP− vs. DC− SP−, see Table 3), with more negative amplitudes for incongruous than congruous duration (see Figure 4). In this time window, there was no interaction with Region or Hemisphere. 

Maximal statistical difference at Talairach coordinates for the significant contrasts detected in the early window are given in Table 4. The source estimation for the semantic effect in the metric task (DC+ SP+ vs. DC+ SP−) showed the maximal statistical difference in the right frontal lobe (BA 6, see Figure 3B). For the duration effect (DC+ SP+ vs. DC− SP+), the right parietal postcentral gyrus (BA 2) reached significance (Figure 3C). In the semantic task, the duration effect (DC+ SP+ vs. DC− SP−) reached the maximal statistical difference in the left sub-lobar insula (BA 13, see Figure 4B). 

### 3.2.2. Second Time Window (1000–1200)

The second time window extended from 1000 to 1200 ms after the word onset (Figure 5). The ANOVA performed on this window showed the main effect of duration (F(1, 19) = 7.65, *p* < 0.012, η_p_^2^ = 0.29) with larger negative amplitude for CD− than CD+. There was also a significant three-way interaction among Task, Duration and Semantic (F(1, 19) = 4.45, *p* < 0.048, η_p_^2^ = 0.19). A simple effect analysis indicated that, for the metric task, there was a significant duration effect (DC+ SP+ vs. DC− SP+, see Table 5), and incongruous duration elicited larger positivity than congruous duration in the predictable sentences (see Figure 5). 

The LAURA source estimation for the duration effect (DC+ SP+ vs. DC− SP+, see Table 6) identified the posterior cingulate located on the left hemisphere as the most active area (BA 23, see Figure 3B).

## 4. Discussion 

This ERP study assessed the neural dynamics underlying the prosodic and the semantic dimensions in Spanish, a free-accent language. Specifically, we wanted to evaluate whether both dimensions share neural process or rely on totally independent neural networks. To this aim, participants listened to sentences in which duration (syllable length) and semantic (predictability) features of the last sentence word were manipulated in the context of two different task demands: metric and semantic judgment tasks. 

In a nutshell, ERP and source estimation results converge in showing statistical interactions between the metric factor duration and the semantic factor predictability. According to the rationale of the study, statistical interactions, especially of electrophysiological measures, support the hypothesis of shared neural networks between some features of prosody and semantics. Additionally, the impact of task demands (metric vs. semantic) is crucial to evaluate to what extent the processes occur automatically or require participants’ full attention. Let us comment on the results of: Behavioral interactions, ERP interactions and overlapping of ERP sources. 

### 4.1. Behavioral

The behavioral results showed interactive effects of both dimensions on the accuracy and response time, which is compatible with the hypothesis of shared neural processes. Thus, the combination of incongruous duration in semantic predictable context or congruent duration in semantic unpredictable context produces the lowest accuracy. In the same vein, the participants spent more time when the target words were unpredictable, independent of duration correctness, or when the duration of the target words was incongruous, independent of predictability. The results evidenced that participants perceive the predictability and duration of the words independently of the judgment task. Importantly, although, unlike the neurophysiological results, the behavioral results did not show interaction between task, semantic and duration, they are compatible with the ERP data, as they reflect different processing stages. The ERP data stage is associated with the on-line processing of the speech signal, and the behavioral data stage involves overall decision-making mechanisms, in which prosodic and semantic values could conflict at the response choice independently of the task. These findings are taken to reflect the interaction between the two dimensions. 

### 4.2. ERPs 

The ERP results identified two components associated with the target word: a relatively early negativity (300–390 ms) and a later extended positivity (1000–1200 ms), both sensitive to the prosodic and the semantic manipulations. The ERP activity clearly showed the effects of duration and semantics in the early time window. Such effects were modulated in a triple interaction between tasks, demonstrating that prosody and semantics processing are closely associated in the brain, and both are modulated by the judgment task requested of the participants. Thus, when participants performed metric judgment tasks, semantically unpredictable sentences elicited larger negative components than semantically predictable sentences but only when the word duration was congruous. In other words, the semantic effect occurs automatically (under metric task), although it was attenuated or disrupted when the word’s prosody was anomalous. Instead, when participants performed the semantic judgment task, incongruous duration words produced a larger negative waveform than congruous duration words but only when the sentences were semantically unpredictable. This suggests that for unpredictable sentences, which require more semantic effort, the role of prosody becomes more conspicuous than in the context of predictable sentences. 

The automaticity of semantic processing in the development of a prosodic task has been supported by prior research using a similar design. For example, Astésano et al. [19] manipulated the fundamental frequency by mixing the beginning of a declarative sentence with the end of a question. Participants had to decide whether the sentences were semantically or prosodically congruous in two different task conditions. In the prosodic task, participants showed the main effect of semantics where the incongruous sentences revealed a larger amplitude than congruous sentences. Additionally, Magne et al. [21] carried out a study in French using a similar design to the present one. They found that when participants focus their attention on the metric task, a semantic effect where semantically incongruous words were larger than congruous words emerged. 

The automaticity of prosody processing when the listener’s attention is focused on semantic features is more controversial. For instance, Astésano et al. [19] did not find statistically significant evidence concerning it, but Magne et al. [21] found a metric congruity effect when participants performed a semantic task, with the metrically incongruous word eliciting a larger N400 component than congruous words. The results of the present study supported the automaticity of prosody processing, but as mentioned above, the triple interaction clarifies the specific contrast in which metric congruity has an effect. Thus, metric differences emerged in the unpredictable sentences, revealing larger N400 amplitudes for modified duration than non-modified duration. Interestingly, this result was not present in Magne et al.’s study [21], likely because these authors changed the penultimate word syllable instead of, as we did in the current study, the first word syllable. This has two implications: first, some studies have suggested that the first syllable is a unit of lexical access both in visual stimuli [50] and auditory stimuli [38] in Spanish; second, this result converges with findings that showed that listeners are more likely to attend to temporal segments of speech that are least able to be predicted [51,52].

These results draw attention to the fact that the semantic effect was evident when the attention is not focused on the predictability feature. Specifically, the contrast between unpredictable sentences and predictable sentences when the word duration was congruous reached significant differences in the metric task, while it did not for the semantic task. This is a novel result relative to prior EEG studies and seems counterintuitive; however, this result reinforces the idea that the semantic processing could be automatic. When the participant’s attention is on the semantic focus, the predictability effect does not emerge “naturally”, but when the attention is on the metric task, it emerges automatically. These results are perhaps related to the manipulation of the semantic dimension that was subtle. The predictable and unpredictable words were sensible in the sentences’ context, so detecting electrophysiological differences between them might require automatic responses more than attentional responses. The participant might focus attention on a different feature, for example, duration, as in this case. These results are different, for example, with respect to the Magne et al. [21] study, which found significant differences between congruent and incongruent words. However, different from our study, theirs included sentences with and without sense. 

Moreover, when participants performed metric judgment tasks, incorrect duration elicited significantly larger negative components than correct duration but only when the sentences were predictable. Interestingly, early window result is related to that obtained in French by Magne et al. [21] who, in a metric task similar to ours, reported that metrically incongruous words elicited a larger negativity in the N400 compared to metrically congruous words. In general, in this task, the correct duration and predictable sentence condition obtained smaller amplitude than the rest of them. The set of these results is interesting because it emphasizes the N400 component as a signature pattern to index mismatch integration in language comprehension. 

Concerning the late positive component of the ERP (LPC), the triple interaction between task, semantics and duration was found again. Namely, while participants performed the metric task, incongruous duration elicited larger positivity than congruous duration in predictable sentences, revealing that the processing of the prosodic feature of the duration continues until 1000–1200 ms after the word onset. Additionally, there were no significant effects when the participant performed the semantic task. These findings point out that this effect is specific to the task demand. These results agree with the effects reported by other studies using prosodic manipulation [19,21]. In these studies, prosodic effects in LPC were found when participants were asked to perform the metric task. Nonetheless, these results disagree with the effects reported by Paulmann and Kotz [53], who utilized real word and pseudoword sentences prosodically expressing six basic emotions by a female and a male speaker. They showed that expectancy violations of emotional prosody and semantics contents elicited a LPC irrespective of the task. 

### 4.3. Sources 

As far as we know, the current study is the first one that estimates the brain sources of ERP components associated with the processing of prosodic and semantic information. Thus, we are reporting the sources of this relationship for the first time. In sum, sources analyses showed sensitivity to duration and semantic predictability at most of the right hemisphere regions for the metric task, and at the left hemisphere for the semantic task.

For the N400 time window, the strongest activity in the metric task occurred mainly at right frontal lobe, specifically the Brodmann area 6. These results agree with studies that suggest that this area could participate in the perception and production abilities [54,55,56,57,58]. Additionally, the present findings are in agreement with Kutas and Federmeier [26], who associated the auditory N400 with a frontal topography. Furthermore, these results agree with previous literature that relates this area to the auditory processing [52,59] and the phonological processing and articulatory speech sounds [26]. In this sense, the present study poses that the premotor area is activated in auditory processing, thus inferring their meaning as indicated by other studies [60,61]. In the semantic task, the area that reached the significant activation was localized to the left sub-lobar insula. The insula (BA13) is related to the lexico-semantic network, and it has been proposed to have a crucial language role by integrating the receptive function [57]. This might be the reason for its activation in the current study, namely, an activation of the semantic network when performing a semantic task. 

The brain area found to be active in the P800 responses in our study was the LPCC (BA 23). The PCC is a highly interconnected hub, which, among other functions, is responsible for regulating sensory attention and cognitive control [62,63]. This is in line with the above results, which point out that the LPC is task dependent. Moreover, apart from the previous data, the source of this area was located on the left hemisphere. This could be because the semantic dimension affected the participant’s attention on duration, as suggested by the statistical interaction obtained at the behavioral and electrophysiological level. Additionally, this result is in line with studies that proposed that the neural network dedicated to the perception of linguistic prosody is bilateral [24]. 

### 4.4. Limitations

This investigation contributes to enriching our understanding about the relationship between prosody and semantics in Spanish, a language with no fixed accent. Yet, further studies will be needed to overcome some drawbacks of this one. This study only assessed how the brain dynamic is sensitive to the duration of the syllables in Spanish words embedded in sentences differing in semantic predictability, whereas other prosodic parameters (pitch, loudness or their combination), pragmatic (emotional prosody, emphatic accent) and semantic features (e.g., emotionality) were not tested. Consequently, a possibility exists that some prosodic and some semantic/pragmatic features of speech could interact at some processing stage. For instance, it has been reported that the incongruence between emotional prosody and emotional semantics enhances the N400 component of the ERP [64].

## 5. Conclusions

To the best of our knowledge, this is the first study of its kind in Spanish. Based on the data obtained, it is possible to conclude that the extraction and integration of the prosodic feature take place automatically in a very short time, about 400 ms from the presentation of the stimuli. The duration and semantic processes are associated, as shown in the behavioral, ERP and source localization analyses. Moreover, the incongruous duration words showed a negative component (N400) or a late positive component larger than congruous duration words. However, more studies are needed to confirm these suggestions.

## Figures and Tables

**Figure 1 brainsci-12-00458-f001:**
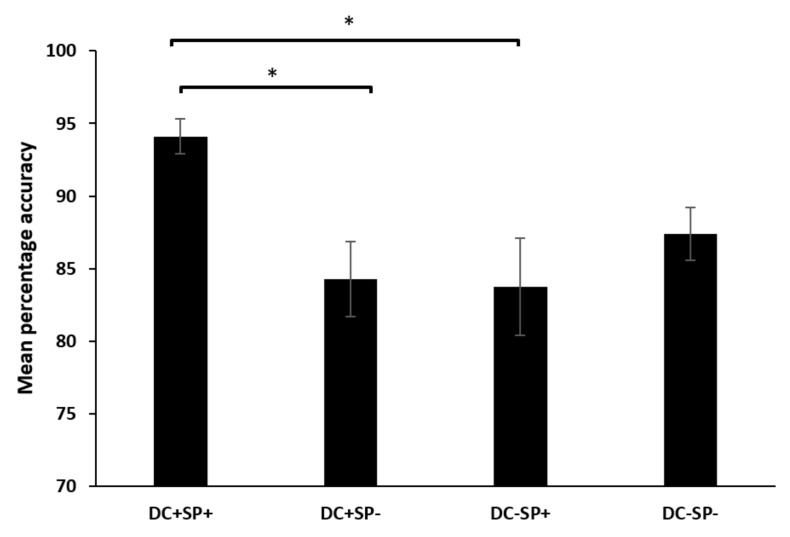
Mean accuracy for the combined duration and semantic factors. Note. DC+: congruous duration; DC−: incongruous duration; SP+: semantically predictable; SP−: semantically unpredictable. The vertical lines correspond to the standard deviations. Asterisk means statistical significance (*p* < 0.05).

**Figure 2 brainsci-12-00458-f002:**
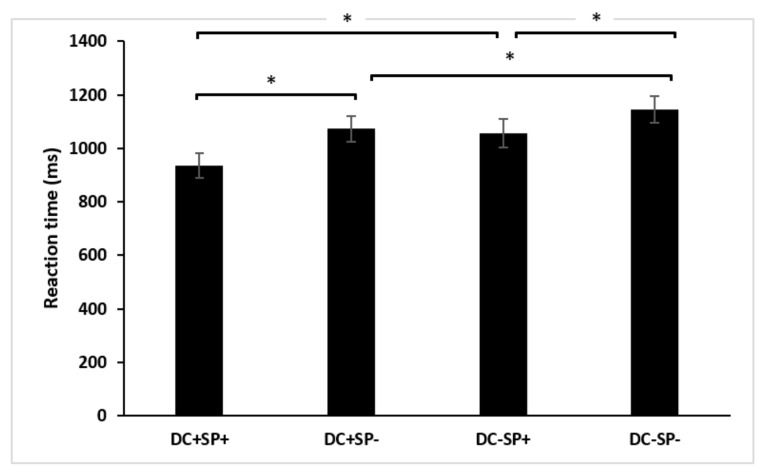
Mean reaction time for the combined metric and semantic dimension. Note. DC+: congruous duration; DC−: incongruous duration; SP+: semantically predictable; SP−: semantically unpredictable. The vertical lines correspond to the standard deviations. Asterisk means statistical significance (*p* < 0.05).

**Figure 3 brainsci-12-00458-f003:**
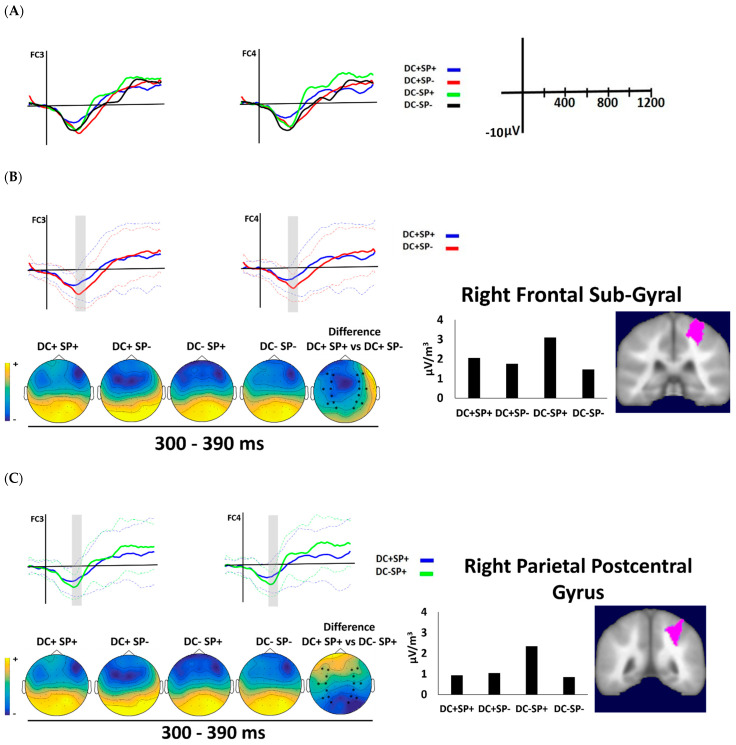
(**A**) Grand average waveforms for congruous duration (DC+) and incongruous duration (DC−) and semantically predictable (SP+) and unpredictable (SP−) sentences for the four experimental conditions at FC3 and FC4 as function of metric task. (**B**) Grand average waveforms of N400 component for DC+ SP+ and DC+ SP− contrast conditions and (**C**) DC+ SP+ and DC− SP+ contrast conditions. The time windows with a contrast are indicated in gray. The dotted lines indicate the standard deviations. Left: spatial distributions resulting from the comparison between the conditions. The asterisks at the topographical maps show the used electrodes. Right: current density values for the contrast and their brain spatial localization for the semantic contrast (rFSG) and duration contrast (rPG).

**Figure 4 brainsci-12-00458-f004:**
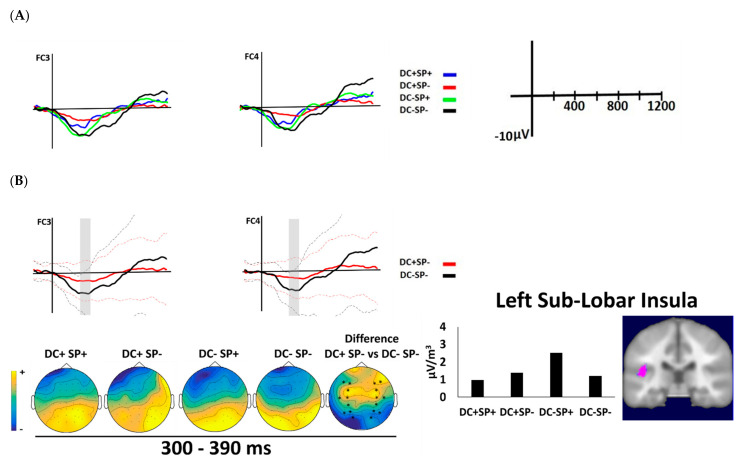
(**A**) Grand average waveforms for congruous duration (DC+) and incongruous duration (DC−) and semantically predictable (SP+) and unpredictable (SP−) sentences for the four experimental conditions at FC3 and FC4 as function of semantic task. (**B**) Grand average waveforms of N400 component for DC+ SP− and DC− SP− contrast conditions. The time windows with a contrast are indicated in gray. The dotted lines indicate the standard deviations. Left: spatial distributions resulting from the comparison between the conditions. The asterisks at the topographical maps show the used electrodes. Right: current density values for the contrast and their brain spatial localization for the duration contrast (L insula).

**Figure 5 brainsci-12-00458-f005:**
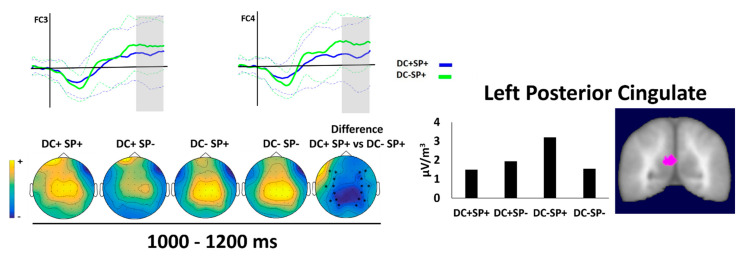
Grand average waveforms of LPC component for DC+ SP+ and DC− SP+ contrast conditions as a function of metric task (see Figure 3A for the four experimental conditions). The time windows with a contrast are indicated in gray. The dotted lines indicate the standard deviations. **Left**: spatial distributions resulting from the comparison between the conditions. The asterisks at the topographical maps show the used electrodes. **Right**: current density values for the contrast and their brain spatial localization for the duration contrast (LPCC).

**Table 1 brainsci-12-00458-t001:** Example of the stimuli in the four experimental conditions.

Duration	Semantic	
	Predictable (SP+)	Unpredictable (SP−)
Congruous (DC+)	El ciclista sufrió mucho en la subida (the cyclist suffered a lot on the climb)	El ciclista sufrió mucho en la visita (the cyclist suffered a lot on the visit)
Incongruous (DC−)	El ciclista sufrió mucho en la **su**bida (the cyclist suffered a lot on the climb)	El ciclista sufrió mucho en la **vi**sita (the cyclist suffered a lot on the visit)

Note. The lengthened syllable is in bold.

**Table 2 brainsci-12-00458-t002:** Mean duration and SD in ms in each syllable in the last word in the four experimental conditions.

Duration		Semantic							
		Predictable (SP+)				Unpredictable (SP−)			
		Syllables			Total Word	Syllables			Total Word
		1	2	3		1	2	3	
Congruous (DC+)	M	129.35	197.70	162.07	489.14	137.32	198.36	185.85	551.75
	DS	22.18	27.61	57.73	63.91	18.52	26.03	59.65	63.31
Incongruous (DC−)	M	202.68	197.70	162.07	582.47	207.74	198.36	185.85	611.97
	DS	24.76	27.61	57.73	64.99	30.68	26.03	59.65	64.83

**Table 3 brainsci-12-00458-t003:** F and significant *p* values from Task, Duration and Semantic interaction for the early window.

N400	Metric Task	Semantic Task
Semantic effect	Contrast: DC+ SP+ vs. DC+ SP− F(1, 19) = 5.30, *p* < 0.033, η_p_^2^ = 0.22	---
Duration Effect	Contrast: DC+ SP+ vs. DC− SP+ F(1, 19) = 5.25, *p* < 0.034, η_p_^2^ = 0.22	Contrast: DC+ SP− vs. DC− SP− F(1, 19) = 6.69, *p* < 0.018, η_p_^2^ = 0.26

Note: Dashed lines means that there was not significant contrast.

**Table 4 brainsci-12-00458-t004:** Maximal statistical difference at Talairach coordinates for the significant contrasts detected in the early window.

N400	Metric Task	Semantic Task
Semantic effect	Contrast: DC+ SP+ vs. DC+ SP− x = 11.82, y = −8.45, z = 28.71	---
Duration Effect	Contrast: DC+ SP+ vs. DC− SP+ x = 15.20, y = −15.20, z = 18.58	Contrast: DC+ SP− vs. DC− SP− x = −21.96, y = −5.07, z = 8.45

Note: Dashed lines means that there was not significant contrast.

**Table 5 brainsci-12-00458-t005:** F and *p* values from Task, Duration and Semantic interaction for the late window.

LPC	Metric Task	Semantic Task
Semantic effect	---	---
Duration effect	Contrast: DC+ SP+ vs. DC− SP+ F(1, 19) = 5.34, *p* < 0.032, η_p_^2^ = 0.22	---

Note: Dashed lines means that there was not significant contrast.

**Table 6 brainsci-12-00458-t006:** Maximal statistical difference at Talairach coordinates for the significant contrasts detected in the late window.

LPC	Metric Task	Semantic Task
Semantic effect	---	---
Duration effect	Contrast: DC+ SP+ vs. DC− SP+ x = −1.69, y = −21.96, z = 11.82	---

Note: Dashed lines means that there was not significant contrast.

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
