# Peer review of "The Brain Dynamics of Syllable Duration and Semantic Predictability in Spanish"

_brainsci, 2022, doi:10.3390/brainsci12040458_

Round 1
Reviewer 1 Report
The manuscript has improved and most of my concerns have been addressed.
Author Response
Dear Reviewer,
We greatly appreciate your commentaries and suggestions on our manuscript. They are very constructive and helpful to improve our work.
We have revised the english language and style.
Thank you.
Reviewer 2 Report
This ERP study investigated prosodic and semantic processing using incongruency judgment tasks. A 2x2 setup was used including prosodically congruent vs. incongruent conditions and semantically congruent vs. incongruent conditions. Two tasks respectively required the subjects to pay attention to and judge the prosodic congruency or semantic congruency of the target words in a sentence context.
The experimental design was appropriate and the behavioral results were properly analyzed and straightforward to interpret. But the ERP results and interpretations were problematic. Here I list the major concerns.
- Term consistency. Table 1 lists the conditions nicely. But in the results section, semantically predictable (SP+) vs. unpredictable (SP-) were sometimes reported as PS+ and PS-. Please be consistent in using the abbreviated terms.
- The introduction talked about lack of scientific literature on the source localization for N400 and LPC responses to motivate hypotheses. I suggest that the authors search the literature again. There are a number of studies that attempted to localize the N400 and LPC components.
- In the "metric" task, the duration contrast (congruent DC+ vs. incongruent DC-) would be expected to elicit the N400 and LPC explicitly, and the semantic processing (predictable SP+ vs. unpredictable SP-) may automatically produce N400 and LPC implicitly. In the "semantic" task, the situation is reversed. Behavioral data showed nice congruency effects in both tasks. But the ERP data, as analyzed and presented, do not appear to support the conclusions and the title. The DC+SP+ sentences should serve as the baseline for comparison to derive the N400 and LPC responses. In each task, there are N400 and LPC components that can be derived from the ERP difference waveforms for duration incongruency (DC-SP+), semantic incongruency (DC+SP-), and duration + semantic inccongruency (DC-SP-). The repeated measures ANOVA analyses need to focus on these N400 and LPC responses from the difference ERP waveforms and see how the explicit vs. implicit task modulates the responses. There were a lot of details in the reported ERP results. Summary tables are needed to help digest the results. For example, the N400 result table can be as follows with the F results and P values reported.
| N400 | Metric Task | Semantic Task |
| Duration (DC-SP+ vs. DC+SP+) | ||
| Semantic (DC+SP- vs. DC+SP+) | ||
| Duration + Semantic (DC-SP- vs. DC+SP+) |
When reporting the source localization results, please show similar tables for the N400 and LPC responses separately.
4. As the brain regions for prosodic and semantic processing looked quite different, the title that emphasizes similarity but not differences for the corresponding N400 and LPC responses does not appear to be justified.
5. In the discussion and conclusions, please highlight the novel contributions of the study. Was there something that had not been reported before or inconsistent with previous findings?
Author Response
Response to Reviewer 2 Comments
Dear Reviewer,
We greatly appreciate your commentaries and suggestions on our manuscript. They are very constructive and helpful to improve our work. Below we provide the point-by-point responses. All modifications in the manuscript have been highlighted in blue.

Round 2
Reviewer 2 Report
The authors have addressed my concerns adequately. I recommend acceptance.
This manuscript is a resubmission of an earlier submission. The following is a list of the peer review reports and author responses from that submission.
Round 1
Reviewer 1 Report
In the current study, the authors investigated the neural network involved in prosody and semantic processing, when listening to a free-accent-language. The authors interpreted their results as a proof of a shared neural network between these two types of processing. In my opinion, this conclusion is not supported by the data, and in general, the question was not properly investigated in this particular experimental design. In fact, the task was not orthogonal to the variables that were manipulated and it is likely that other brain mechanisms, such as attention and decision making, modulated brain responses.
Introduction
Overall the introduction needs an extensive revision: because of the quality of writing; because some studies should be better described to make them more understandable for naïve readers; and because the aim of the study and the expected results are not clearly stated
- First paragraph: “multiple acoustic parameters such as F0, duration, and intensity whose perceptual correlates are pitch, length, and loudness, respectively; all of which contribute to the perception of the suprasegmental structure of sentences” maybe the first time change F0 with frequency (fundamental frequency). Also length is not defined as the perceptual correlate of duration
- It would be nice to explain the concept of prominence for naïve readers
- In duration approaches: “Behavioral studies have shown that durational characteristics” these are rather neuropsychological studies since are conducted on patients with brain damage
- “confirming that the left hemisphere is biased to process temporal cues, and the right hemisphere is biased to process pitch cues” this is not a bias, but rather a lateralization of functions/hemispheric specialization
- The authors referred to the N325 as a “larger negativity”, but maybe this electric potential could be introduced better. For instance, explaining that this is a negative brain response, probably related to stress discrimination, peaking around 325 ms post-stimulus onset and maximum at the frontocentral scalp
- The study by Bohn et al. is not well introduced and hard to understand without checking the reference
- The present study: in the first paragraph the authors described the aim of the study and the variable that they manipulated. I found this explanation not exhaustive considering that the authors did not make explicit hypothesis on the expected neurophysiological results (what ERP components are considered and why these components should be modulated when listening to a free-accent-language as compared to fixed-accent language)
- All the expected results (concerning ERPs and source localization) are not clear
Methods
- ERP studies usually adopt a paradigm where brain responses are measured while participants are exposed to a series of stimuli and to ensure participants attention, subjects are required to perform a task on target trials that are usually discarded from the analysis because within these trials, brain responses might contain artifacts generated by the task, rather than by pure perceptual processing. In this study, participants were asked to decide if the last word of each sentence was well pronounced or not, paying attention to either the metric or the semantic, and had to respond by pressing a button. Because the task is not orthogonal to the variable that has been manipulated, there could be an effect of the task on brain responses overlapped to the mechanism of interested (i.e. semantic/duration processing).
- On average, what was the duration of the last word in the sentence? Were the words all similar in overall length?
- Were the data averaged across electrodes within the same region (occipital, parietal…)?
- How long were the epochs?
- Why did the authors decide 200 ms for the baseline? What was the temporal interval between words in the sentence? Was the voice still uttering the previous word in these 200 ms of baseline?
- I don’t understand the logic behind the analysis. Again it seems that there isn’t a clear hypothesis about ERP components and localization of brain responses and that the authors considered all the electrodes in a more explorative, data-driven, approach
- “Mean amplitudes of significant consecutive time windows were then averaged and entered into a second ANOVA” did the authors use a minimum number of consecutive significant time windows as a thresholds for inclusion?
- The authors described the presence of 3 factors: task, duration congruence and semantic predictability. In this way, participants sometime were performing a semantic task when duration was manipulated and vice versa. Is that correct? I think that this part is not explained clearly in the manuscript. What is the reason behind the choice of this design?
Results
Behavioral results:
- How was accuracy calculated?
- The statistical analysis are reported in a very confusing way. In the accuracy analysis, I understand that there is a main effect of semantic and duration, but not task. But are there significant interactions between factors?
EEG results:
- “The first-time window was a negative going waveform, extending from 300 to 390. In the lateral electrodes” which lateral electrodes?
- Figure 3: I find it more helpful to have ticks on the graph to better understand the timing and the voltage of brain responses. Likewise it would be helpful to report the voltage range of the topographic plot and explain weather each headplot represents the instantaneous amplitude at a certain latency or rather the average amplitude in a temporal window. Maybe assign a title to the subsection of the figure (a, b and c) to make it more graspable. There are few typos in the figure legend: DC+DP- (instead of DC+SP-) in figure 3a and c.
Discussion
- The authors stated that the idea behind the study was to evaluate whether prosody and semantic processing share neural processes or rely on totally independent neural networks and conclude that the results support the hypothesis of shared neural networks between some features of prosody and semantics. However, as mentioned above, the task was not orthogonal to the main variables (i.e. duration and semantic) and, therefore, it is almost impossible to make conclusions about neural networks, considering that the two tasks might recruit different areas more specialized to either semantic or duration processing but also areas involved in decision making, and this activity overlaps with the detection of incongruencies (wrong syllable duration/wrong word) that likewise requires a broad range of processes and involves a an extended network.
- The effect on the behavioral responses are discussed as an evidence of shared neural mechanisms, but changes in accuracy/RT related to incongruency and prediction errors can also be simply explained by cognitive load and attentional interferences
Some of the typos:
- Prosody is a complex aspect of communicative speech acts (should be act)
- the most studied parameter has been F0, which, according to some authors, (it) plays the principal role to mark prominence in speech
- The study of duration has been carried out (maybe change with “the relevance of duration features in prosody has been investigated throughout different approaches…”)
- durational characteristics of prosody distinguish meanings (maybe change with “duration features in prosody can be used to distinguish lexical items”)
- is a useful tool to addressing questions (either “to address” or “for addressing”)
- and 1000 ms after the presentation of the cross, the subject listened the sentence (listened to the sentence)
- To estimate the cortical regions involved in the semantic and metric effect, we analyze(d) the brain activity in the early (300-390 ms after onset of the target word) and in the late time window (800-900 ms)
Reviewer 2 Report
The authors conducted a careful study of duration in speech, and report a novel paradigm.
When introducing the topic of ERPs and reviewing the literature on semantics and syntax, the authors should cite more recent, relevant work: Morgan et al. (2020).
Morgan, E.U.; van der Meer, A.; Vulchanova, M.; Blasi, D.E.; Baggio, G. Meaning before grammar: a review of ERP experiments on the neurodevelopmental origins of semantic processing. Psychonomic Bulletin & Review 2020, 27, 441-464.
The results are presented beautifully, with excellent figure design and structure.
Reviewer 3 Report
This ERP study examined effects of syllable duration and semantic predictability manipulation in Spanish words requesting listeners to judge prosodic or semantic congruency in a sentence context. The experimental design allows not only explicit congruency effects comparison of prosodic vs. semantic processing but also implicit (or automatic) processing of these two informational dimensions as the semantic congruency judgement may involve prosodic incongruency and the prosodic congruency judgement may involve semantic incongruency in the 2(task)x2(syllable duration)x2(semantic predictability) factorial design. Behavioral accuracy and reaction time data showed significant interaction effects between duration and semantics factors. Accuracy was highest and reaction time was the shortest when the syllable duration was congruent and the target word was semantically predictable. Accuracy was significantly reduced when involving both incongruency in duration and unpredictability in semantics. Further reduction in accuracy occurred when involving conflicting information in one informational dimension but not the other. Reaction time data showed that it took the listeners longest time to judge when both duration incongruency and semantic unpredictability were involved. These data showed some different interaction effects in the two measures in addition to revealing how duration information and semantic predictability influence each other in the judgement. The ERP data showed N400 responses for both tasks and a later P800 for prosodic processing regardless of explicit or implicit instructions. Source localization of the ERP components revealed different cortical origins for the semantic N400 and prosodic N400 responses. P800 response also showed different locations depending on the task.
Overall, this study investigated an important research topic and the findings make a timely contribution to the existing literature. There are some theoretical and technical issues that require further clarification.
- In the introduction, it is problematic to restrict the interpretation of N400 to semantic processing as it could be elicited by informational conflicts involving semantic congruity, phonetic congruity, and affective/prosodic congruity as well as contextual predictability. There could be multiple N400 source generators for conceptual/lexical (even phonetic) integration conflicts depending on the task.
- The term P800 showed up in a sudden in the outcome predictions. Please provide some background information with references that this P800 reflects the late positive component for processing prosodic incongruency.
- The method section provided detailed descriptions of the materials, test procedures, and analysis. Some clarifications are needed. (a) In preparing the sentences, was intensity normalization used? What sound pressure level was used in presenting the stimuli? (b) The phrase "re-referenced offline to mastoid electrodes" is unclear. I assume it is the average of the M1 and M2 electrodes. (c) The time window comparison by averaging every 10 ms consecutively does not appear to be a standard procedure. Fieldtrip has implemented the permutation function to help determine time points where significant differences between two ERP waveforms occur. Can you confirm the choice of the 300-390 ms and 800-900 ms for the target components with the permutation analysis?
- As the accuracy rates are quite different among the conditions, did the ERP averaging only include the correct response trials?
- As the duration manipulation lengthened the first syllable, it could prolong/delay the late positive component response for processing prosodic incongruency in the ERP subtraction.
- In the ERP waveforms data (Figure 3), there is a typo. The legend DC+DP- should be DC+SP-.
- In discussion, please note that N400 and LPC responses can even be elicited with nonsense syllables involving cross-modal phonetic/emotional prosodic information conflict (https://doi.org/10.1016/j.neuropsychologia.2016.01.019 https://pubs.asha.org/doi/10.1044/2020_JSLHR-19-00329) Thus N400 and LPC can be elicited by general information integration conflicts other than semantic incongruity or unpredictability.
- Please reconsider the wording of "automatic processing" in the explicit vs. implicit instruction comparison, especially given that N400 can be elicited by informational integration conflict without involving semantic processing at all. The interesting point is whether the implicit semantic unpredictability boosted the prosodic N400 more than the other way around or their mutual influences were of comparable size.
- The LAURA method is a distributed source estimate method. It is confusing to refer to the source location results as "dipole" location since the later term generally refers to fixed/moving dipole localization methods.
- Please move the limitations section ahead of conclusions.
Round 2
Reviewer 1 Report
Overall the manuscript has been improved, but I still have few concerns, mostly about the interpretation of the ERPs results.
- If I understood correctly, for each task, each experimental condition was characterized by 28 total trials (considering that participants performed two blocks, each containing 14 trials for condition), is that right?. If yes, this is a very limited amount of trials for an EEG experiment, especially considering that some trials can be removed because of noise and/or motor artifacts. Is it possible to plot the standard error of the ERP components?
- Behavioral results, Figure 1: perhaps it would be better to change “Mean percentage accuracy” with “mean accuracy (% correct responses)”, assuming that this is how accuracy was calculated
- Metric task, N400 effect: “Follow-up contrast revealed that in the metric task there was a modulation of semantics; that is, unpredictable sentences elicited a more negative waveform (N400) than predictable sentences but only in the correct duration condition”. From the figure it doesn’t seem that the N400 is more negative in the DC+SP- condition, but rather that the amplitude of the N400 in this last condition is similar to the others, while the DC+SP+ condition is smaller. Basically, the N400 appears to be sensitive to any type of incongruency, in this task
- Semantic task, N400 effect: “By contrast, in the semantic task, duration effect only happened for the unpredictable sentences, with more negative amplitudes for incongruous than congruous duration”. To be precise, in this task the N400 seems to be sensitive only to the maximum incongruency (DC-SP-). How do the author discuss this task modulation?
- “The amplitude of incongruous duration words was larger than that of congruous duration words. In the same vein, the amplitude of unpredictable words was greater than predictable ones”. That’s not entirely true, for instance in the semantic task the amplitude DC+SP- is the smallest one. How do the author explain that this condition isn’t significantly different (and more negative) than DC+SP+?
Minor comments on the Introduction:
- “The event related potential technique (ERP) has found a signature pattern of brain activity to index mismatch integration (e.g. semantic or phonetic congruity)” The term mismatch integration is not entirely correct, better to just say semantic/phonetic congruency
- “N400 is one of the most important language research component which peak around 400 ms. Specifically, it is a useful for addressing questions on the integration of prosodic information in auditory processing [20], providing a measure of the time course of prosodic integration in semantic [22,23] and syntactic processes [20, 27]” N400 is one of the most important ERP component……it is a useful (tool?)
Author Response
Dear Reviewer,
We greatly appreciate your commentaries and suggestions on our manuscript. They are very constructive and helpful to improve our work. Below we provide the point-by-point responses. All modifications in the manuscript have been highlighted in red.

Reviewer 3 Report
While the authors made efforts to address the concerns, I still see a number of problems and errors with the revised version.
- There are procedural errors with the analysis. The researchers averaged all the ERP responses for correct and incorrect trials based on their experimental conditions. However, brain responses for correct and incorrect trials are known to be quite different and should be separated in analysis. For instance, the incorrect trials may have no noticeable N400 responses in a congruency judgment task. Some examples are provided here.
Batterink, L., Karns, C. M., Yamada, Y., & Neville, H. (2010). The Role of Awareness in Semantic and Syntactic Processing: An ERP Attentional Blink Study. J Cogn Neurosci, 22(11), 2514-2529. doi:10.1162/jocn.2009.21361
Silva-Pereyra, J., Gutierrez-Sigut, E. and Carreiras, M. (2012), ERP study of coreference in Spanish: Semantic and grammatical gender cues. Psychophysiology, 49, 1401-1411. https://doi.org/10.1111/j.1469-8986.2012.01446.x
- Figure 1 shows the DC+SP+ condition with much higher accuracy rate than the other conditions. Figure 2 shows this easiest condition with the shortest reaction time. How would the ERP data be influenced by the levels of task difficulty for the different stimulus conditions?
- The exploratory analysis for selecting the time windows appears to have missed the time points which show maximal difference between the conditions of interest as plotted in the ERP waveforms in Figure 3.
- Figure 3 is confusing and has some errors (including typos; for instance, the label "difference" was partially covered up in Figure 3C). It is confusing why for each contrast of interest, the localization plots in Figure 3 have only one graph on the right but the ERP waveforms and current density magnitude data show 4 conditions on the left. I mean, there should be multiple comparisons among the 4 conditions for the congruency effects and implicit vs. explicit instruction in the 2x2 design. For instance, the contrasts for the implicit and explicit task instructions need to be separated and compared in t-maps for source localization if the authors want to make claims about automatic vs. effortful processing of semantic and prosodic information. It is also unclear to me why the N400 and LPCs are merged for the semantic task in Figure 3C but these two components were plotted separately in Figures 3A and 3B for the metric task. For each task, there would be congruency contrasts in prosody and semantics and explicit vs. implicit processing. The differences for those should follow the 2x2 design to reveal the full picture when plotting the results.
- The unit of measurement in the method section states the localization data to be in "ampere per square millimeter". But the localization plots use units in microampere per cubic meter in Figure 3. That does not seem to match up.
- There are a number of typos and unclear/imprecise descriptions throughout the manuscript. For instance, the first word for Figure 3 caption should be "Grand" not "Gran". The fifth line of Figure 3 caption states "the used electrodes", but it is unclear what they are used for. Are these the electrodes selected for statistical analysis?
- The authors claimed in their response that the duration difference in the first syllable did not make a difference in the LPC result. But I do not think that is correct. From what I can see in Figure 3B, the LPC effect appeared to be much earlier and stronger when the duration factor was controlled in the DC-Sp+ vs DC-SP- comparison than the LPC in the other comparison (DC+SP+ vs. DC-SP+). It should be relatively easy to subtract the conditions and plot the ERP difference waveforms to show when the peak differences actually occur in each contrast of interest. If my observation is correct, then the analysis window of 800-900 ms would also be problematic for the DC-Sp+ vs DC-SP- comparison.
Author Response

(The authors gave the same response as above.)
